# Microbiological Quality of Raw Donkey Milk from Serbia and Its Antibacterial Properties at Pre-Cooling Temperature

**DOI:** 10.3390/ani13030327

**Published:** 2023-01-17

**Authors:** Ljubiša Šarić, Tamara Premović, Bojana Šarić, Ivana Čabarkapa, Olja Todorić, Jelena Miljanić, Jasmina Lazarević, Nedjeljko Karabasil

**Affiliations:** 1Institute of Food Technology, University of Novi Sad, Bulevar Cara Lazara 1, 21000 Novi Sad, Serbia; 2Faculty of Information Technology and Engineering, Union-Nikola Tesla University, Staro Sajmište 29, 11000 Belgrade, Serbia; 3Faculty of Veterinary Medicine, University of Belgrade, Bulevar Oslobođenja 18, 11000 Belgrade, Serbia

**Keywords:** donkey milk, microbiological quality, antibacterial activity

## Abstract

**Simple Summary:**

The purpose of this study was to investigate the microbiological quality of raw Domestic Balkan donkey milk and to determine changes in its microbiota during six-day storage at 4 °C, and to investigate the antibacterial activity of this milk toward some foodborne pathogens at a selected pre-cooling temperature of the milk (15 °C). Lysozyme and lactoferrin content in donkey milk were also determined. Absence of pathogens and low levels of total bacterial count, lactic acid bacteria and coagulase-positive staphylococci in 137 samples tested indicated good hygiene practice during milking. Regarding the antimicrobial potential of donkey milk, *E. coli* appeared to be the most sensitive to the antibacterial potential of donkey milk while *S. aureus* was the most resistant to the antibacterial effect of this milk. The microbial quality of donkey milk during prolonged storage also indicated the antibacterial activity of this milk. Lysozyme and lactoferrin were present at concentrations of 2.63 ± 0.03 g L^−1^ and 15.48 mg L^−1^, respectively.

**Abstract:**

The aim of this study was to examine the microbiological quality of raw donkey milk of an indigenous Serbian breed as well as the changes in the microbial populations during storage at 4 °C. In addition, antibacterial activity of donkey milk against *E. coli*, *L. monocytogenes* and *S. aureus* at 15 °C as well as the content of the two main antibacterial proteins lysozyme and lactoferrin were investigated. Microbiological examination of 137 individual milk samples collected over a period of 21 months showed good microbiological quality since foodborne pathogens such as *Salmonella* spp. and *L. monocytogenes* were not detected in any of the analyzed samples, while the number of *E. coli*, Enterobacteriaceae, total coliform bacteria, sulfite-reducing Clostridia and aerobic sporogenic bacteria was below the limit of quantification (<1 cfu mL^−1^). During the six-days storage at 4 °C, total bacterial counts and the counts of lactic acid bacteria remained at the initial level while pathogenic bacteria were not detected. The strongest antibacterial activity of the tested milk was observed against *E. coli*, while *S. aureus* was the least sensitive to milk antibacterial compounds. Although further research is needed to fully elucidate the antibacterial mechanism and synergistic activity of different compounds in donkey milk, the high content lysozyme (2.63 ± 0.03 g L^−1^) and lactoferrin (15.48 mg L^−1^) observed in tested milk could contribute to its strong antibacterial activity and extension of the storage period during which it can be safely consumed.

## 1. Introduction

Cow’s milk has the greatest importance in global milk production, but the production of non-cow’s milk at global level has been growing steadily over the last 50 years [1]. In developing countries, intensive consumption of non-cow milk is the result of local tradition and economy, while in developed countries it is a consequence of popularization of a healthy diet. Although there are no reliable statistical data on the share of donkey milk (DM) in global milk production, it is believed that less than 0.1% of global milk production is accounted for by DM and equine milk [1].

In recent years, DM has become the subject of scientific investigations, although it has been part of traditional diets and medicines worldwide since ancient times [2,3,4,5,6]. The results of pediatric and clinical studies have indicated this milk as a good alternative for children suffering from cow’s milk protein allergy, which has led to its popularization [5,7,8,9,10,11,12]. Further studies have shown various aspects of its functionality such as antioxidant, antibacterial, antiviral, anti-inflammatory, antiproliferative, and anticancer activity [5,10,12,13,14,15,16,17,18,19]. DM has beneficial effects in the prevention of atherosclerosis [5,14] and cardiovascular diseases [5], as well as antidiabetic and immunomodulatory properties [5,10,19,20]. In recent years, DM is gaining importance and international acceptance as a “health food” and “pharma food” [5,19] and is considered a “niche business” with high commercial value [21]. Accordingly, demand for DM is constantly increasing, as is the number of commercial dairy donkey’s farms [12]. Although processed DM (pasteurized, UHT-treated, freeze-dried, and powdered) safe for human consumption can be found in the market of developed countries (stores, pharmacies, online sales) [5,10,11,22,23,24], DM is still mainly produced in rural areas of Asia, Africa, Eastern and Southern Europe, where this milk is traditionally consumed raw [25,26,27,28]. In the Balkan region, raw DM was traditionally used to treat various respiratory diseases [18,29]. Even today, when the risks associated with the consumption of raw milk are well known [5,30], DM is still consumed raw in Serbia due to strong beliefs that heat treatment reduces the functionality of milk. Although there is no reliable evidence of the negative effect of heat treatment on the functionality of DM, there are a few reports on the adverse effect of heating on some valuable components (α-tocopherol, vitamin C, nitrogen profile, lysozyme, lactoferrin, unsaturated fatty acids) of DM [5,10,22,31,32,33].

Different views on the health benefits of consuming raw milk are reflected in national regulations. For example, the sale of raw milk for direct human consumption is strictly prohibited in Canada and Australia, while it is allowed in 29 US states under certain conditions [34]. On the other hand, the EU does not recognize the sale of raw milk for direct human consumption as illegal and, through Regulation EC 853/2004 [35], allows each Member State to regulate this sale on its territory through its national legislators. These member states must maintain or introduce appropriate health measures to ensure the achievement of the objectives of this regulation in their territory. The Regulations on the Quality of Raw Milk [36] allowed the sale of raw DM in the territory of the Republic of Serbia, as raw milk is defined as milk produced by the secretion of the mammary gland, not only from conventional dairy cattle (cows, goats or sheep) but also other domestic animals (mares, donkeys, buffaloes). *Official Gazette of the Republic of Serbia* 111/2017 [37] contains regulations determining the quantities of primary products, including raw milk, used to supply consumers. According to this regulation, primary producers must ensure that raw milk is marketed directly to the final consumer no later than 24 h after the milk has been milked and cooled to 4 °C.

Although DM contains numerous antimicrobial agents such as lysozyme (LZ) and lactoferrin (LF), which play an important role in protecting milk, its valuable nutritional content still provides a favorable environment for the growth of microorganisms including various pathogens [5,30,38].

Considering that consumers in Serbia traditionally consume raw DM, this study had three objectives: (i) to monitor the microbiological quality of raw DM of an indigenous Serbian breed over a period of 21 months; (ii) to determine the microbiological quality of raw DM during 6-day storage at 4 °C; (iii) to investigate the antibacterial activity of raw DM against some foodborne pathogens at a selected pre-cooling temperature of the milk (15 °C).

## 2. Materials and Methods

### 2.1. Sample Collection

Over a period of 21 months, 137 individual milk samples were collected from 33 multiparous Domestic Balkan donkeys. Morning hand milking was carried out monthly or bi-monthly in the special Zasavica nature reserve, Serbia. The range of repeated sampling per individual animal was between 3 and 8. The milking animals, aged 3 to 7 years, were in good physical condition and showed no signs of disease or unusual behavior. The donkeys participating in the experiment were in different lactation periods (75–210 days postpartum). Every morning before milking, the udders were washed under cold running water and then dried with a towel. The first squirts of milk were discarded. The individual milk samples were collected in sterile bottles and stored in an ice box at 4 °C during transport to the laboratory. Milk samples were analyzed immediately upon receipt at the laboratory.

### 2.2. Determination of the Microbiological Quality of Donkey’s Milk

Microbiological examinations of the DM samples were carried out by enumeration of total bacterial count [39], yeasts and molds [40], coagulase positive staphylococci [41], beta-glucuronidase-positive *Escherichia coli* [42], Enterobacteriaceae [43], coliform bacteria [44], sulfite-reducing Clostridia [45] and *Bacillus cereus* [46]. Aerobic sporogenic bacteria were enumerated by incubating previously heated (100 °C, 5 min) milk samples on nutrient agar (Himedia, Mumbai, India) at 30 ± 1 °C for 72 h according to the respective ISO methodology. The counts of lactic acid bacteria were determined by incubation (30 ± 1 °C, 72 h) of inoculated Man, Rogosa and Sharpe (MRS) agar (LabM, Bury, UK) according to ISO 15214:1998. Detection of *Salmonella* spp. and *Listeria monocytogenes* was performed according to ISO methods [47,48]. All experiments were performed in triplicate.

### 2.3. Determination of the Changes in the Microbial Population of Donkey’s Milk during Storage

The bulk DM prepared by mixing nine randomly selected individual milk samples was homogenized and divided into three subsamples and used to determine the changes in the microbial population of DM during the six-day storage at 4 °C. These changes were monitored by enumerating the total counts of bacteria, yeasts and molds, coagulase-positive staphylococci, beta-glucuronidase-positive *Escherichia coli*, aerobic sporogenic bacteria, lactic acid bacteria, Enterobacteriaceae and coliform bacteria using the above-mentioned methods. The detection of *Salmonella* spp. and *L. monocytogenes* as well as the enumeration of *Clostridium perfringens* were performed according to ISO methods [47,48,49].

### 2.4. Determination of the Content of Lysozyme and Lactoferrin

The content of LZ and LF was determined in the bulk DM used for the antibacterial test. A modified method from Tidona et al. [14] was used to prepare the samples. After dilution of milk samples in buffer (0.125 M Tris-HCl, 4% SDS, 2% glycerol, 2%. β-mercapto-ethanol, pH 6.8) in a 1:1.5 (*v*/*v*) sample: buffer ratio, these milk dilutions were heated at 100 °C for five minutes. For the chip-based separations, the Agilent 2100 Bioanalyzer (Agilent Technologies, Santa Clara, CA, USA) was used in combination with the Protein80 Plus LabChip Kit and the dedicated Protein 80 Software Assay on 2100 Expert software. The Protein 80 LabChip kit served as the protocol for preparing the chips. According to the convention for SDS-PAGE [50], fractionation is size-based and the profiles show the smallest proteins appearing first in the profiles but at the bottom of the gel patterns. Bovine serum albumin was used as a standard for quantification of milk proteins. All samples were analyzed in triplicate.

### 2.5. Antibacterial Assay

#### 2.5.1. Preparation of Inoculum

The antibacterial activity of the DM was tested against *E. coli* ATCC 8739, *S. aureus* ATCC 25923 and *L. monocytogenes* ATCC 19111. After overnight incubation on nutrient agar at 37 ± 1 °C, well-isolated colonies of each tested bacteria were selected and transferred with an inoculating loop into a tube containing sterile saline and mixed thoroughly. A DEN-1 densitometer (Biosan, Riga, Latvia) was used to prepare the bacterial suspension whose turbidity was equal to the 0.5 McFarland standard. Ten-fold sequential dilutions of the bacterial suspensions were made in 0.1% peptone saline.

#### 2.5.2. Test at 15 °C

The bulk DM was homogenized and divided into three subsamples (25 mL each) and used for the antibacterial test. These subsamples were artificially contaminated with the selected bacterial strains using appropriate dilutions of the bacterial suspensions. The contamination level was 4.5 log cfu mL^−1^. Contaminated samples were stored at 15 °C for 96 h. Changes in the number of tested bacteria were monitored every 24 h. Three independent antibacterial assays were performed for each selected bacterial strain. *S. aureus*, *E. coli*, and *L. monocytogenes* were enumerated according to ISO methods [41,42,51]. Nutrient broth (Himedia, Mumbai, India) inoculated with 4.5 log cfu mL^−1^ of the test bacteria was used as a positive control, while non-inoculated DM served as a negative control. All experiments were performed in triplicate.

### 2.6. Statistical Analysis

Results were expressed as means ± standard deviations of triplicate analyses for all measurements. Analysis of variance was performed by Duncan’s multiple comparison tests using STATISTICA version 10 (StatSoft Inc., Tulsa, OK, USA). *p*-values < 0.05 were considered significant.

## 3. Results

### 3.1. Microbiological Quality of Donkey’s Milk

*Salmonella* spp. and *L. monocytogenes* were not detected in any of the 137 samples of raw DM examined. The number of *E. coli*, Enterobacteriaceae, total coliform bacteria, sulfite-reducing Clostridia and aerobic sporogenic bacteria was below the limit of quantification (<1 cfu mL^−1^) in all samples tested, as was the number of *Bacillus cereus* and yeasts and molds (<10 cfu mL^−1^). Proportion of DM samples with different number of total bacterial count (TBC), lactic acid bacteria (LAB) and coagulase-positive staphylococci (CPS) in the total number of 137 samples tested is listed in Figure 1.

Although the TBCs in the raw DM varied from 0.8 ± 0.1 to 4.6 ± 0.02 log cfu mL^−1^, they did not exceed 2.5 log cfu mL^−1^ in 85.41% of the samples tested. In addition, the TBC was less than 1.0 log cfu mL^−1^ in five samples (Figure 1). The results obtained showed that 82.48% of the total samples examined had LAB counts of less than 1.5 log cfu mL^−1^ (Figure 1). The highest count of these bacteria was 2.55 ± 0.05 log cfu mL^−1^. A total of 51 samples contained CPS ranging from 1.0 ± 0.0 to 2.62 ± 0.02 log cfu mL^−1^ (Figure 1). Although the number of CPS was generally proportional to the TBCs, these bacteria were not detected in milk samples with TBCs above 4 log cfu mL^−1^.

### 3.2. Changes in the Microbial Population during Storage

TBC and LAB were not significantly different on day 0 and day 6 (*p* > 0.05). Aerobic sporogenic bacteria, Enterobacteriaceae and coliforms were below the limit of quantification until the end of storage (Table 1), as were coagulase-positive staphylococci, *E. coli*, *B. cereus*, sulfite-reducing Clostridia and yeasts and molds. *Salmonella* spp. and *L. monocytogenes* were not detected in DM during the storage period.

### 3.3. Characterisation of the Donkey Milk Proteins

The gel image of DM is shown in Figure 2. LZ and LF were identified as proteins with molecular weights of 15 and 78.1 kDa, respectively [14].

The levels of LZ and LF in bulk DM (used to determine changes in the microbial population during milk storage and for the antibacterial test) were 2.63 ± 0.03 g L^−1^ and 15.48 mg L^−1^, respectively.

### 3.4. Antibacterial Activity of Donkey Milk

The results of the antibacterial test performed on selected foodborne pathogens are shown in Figure 3. DM showed strong antibacterial potential against the tested bacteria, as the numbers of all three pathogens were significantly lower (*p* < 0.05) in DM compared to the positive controls (Figure 3).

The antibacterial activity of DM against *L. monocytogenes* can be characterized as growth inhibitory due to the prolongation of the lag phase during the first two days of storage. After this period, growth of *L. monocytogenes* was recorded in DM, but it was significantly slower (*p* < 0.05) compared to growth in nutrient broth, where the number of this bacterium increased from the beginning of storage (Figure 3a).

The growth rates of *S. aureus* in DM and in nutrient broth were similar on the first day of storage at 15 °C (Figure 3c). This was obviously a growth lag phase in which the bacteria adapted to the changed environmental conditions. In the nutrient broth, the number of *S. aureus* increased sharply from day 1 to day 2, while in DM a significantly slower (*p* < 0.05) growth of this bacterium was observed. This indicates that the antimicrobial compounds in DM were still active. The intense growth of *S. aureus* after the second day of storage indicates that DM had no remaining antibacterial activity, especially after the third day of storage when the growth curves in both matrices were almost the same (Figure 3c).

Of all the bacterial species tested, DM showed the strongest antibacterial activity against *E. coli* (*p* < 0.05), as the number of this pathogen decreased from the beginning of storage and reached the lowest value (*p* < 0.05) on day 3 (Figure 3b). Thereafter, the number of *E. coli* began to increase, but very slowly, and remained significantly lower (*p* < 0.05) compared to its initial number.

## 4. Discussion

### 4.1. Microbiological Quality of Donkey’s Milk

In general, the sources of microbiological contamination of raw milk can be the air, udder, milking equipment, feed, soil, feces and grass [52,53]. The microbiological profile of raw DM is primarily influenced by animal health, raw milk management, animal hygiene management and dairy management practices on donkey farms [54]. Furthermore, ambient temperature, season, storage conditions, staff health and cleanliness can influence the load and type of microorganisms in milk [52].

The previously reported TBC in raw DM varies from 1.08 to 5.77 log cfu mL^−1^ [2,10,12,28,29,55,56,57,58,59,60,61,62]. Salimei and Chiofalo [63] showed that the high TBC determined in raw DM was associated with hygiene deficiencies during machine milking. The TBC content in raw DM from farms practicing machine milking ranged from 4.05 to 5.77 log cfu mL^−1^, while the TBC content in DM samples from farms practicing manual milking was as high as 2.84 log cfu mL^−1^ [61]. Accordingly, other authors reported an overall poorer microbiological quality of machine milked DM compared to hand milked DM [64]. The use of mechanical milking requires effective hygiene for the milking equipment. Otherwise, the equipment can become a major source of contamination of the milk, especially with microorganisms resistant to cleaning and disinfection [32]. In this study, the milk samples were taken individually and not mixed afterwards, which could also affect the microbiological quality. Our results are in agreement with the results of a previous study conducted on 101 individual DM samples where TBC did not exceed 2.40 log cfu mL^−1^ [58]. Only six milk samples (4.38%) tested in this study had a TBC level above 4 log cfu mL^−1^ (Figure 1), probably due to hygiene failures during milking.

A wide range of LAB counts (1–4.24 log cfu mL^−1^) in raw DM has been reported in the literature [2,10,15,28,56,63,65]. Given that LAB represents the natural indigenous microbiota of raw milk [66], the varying burden and types of LAB in DM can be attributed to several factors, including breed-specific characteristics, differences in the content of individual milk components, diet, environment, maintenance, season, health status, and the method used for identification [12,67].

As the CPS was detected in low numbers (Figure 1), these pathogens were not capable of producing large amounts of enterotoxins required to cause foodborne poisoning. Therefore, staphylococcal enterotoxins were not investigated in this study. According to previous reports, the content of CPS in raw DM from Italy, Greece, Cyprus, Turkey and Serbia had values up to 2.26 log cfu mL^−1^ [11,12,15,62]. The CPS in raw milk may originate directly from the skin of milking animals and workers or arise as mastitis microorganisms [68]. Therefore, regular and adequate cleaning and disinfection of the milking equipment, the udders of the animals and the hands of the workers is a prerequisite for obtaining raw milk free of CPS and staphylococcal toxins [68]. In addition, regular monitoring of milking animals for mastitis is necessary, as *Staphylococcus* spp. has been shown to cause subclinical mastitis in milking animals [69].

Although milk is generally a suitable environment for the growth of *Salmonella* spp. and *L. monocytogenes*, the presence of these pathogens in raw DM has also not been detected in previous studies by other authors [10,11,28,59,60,61,62,64,70,71].

According to the available literature sources, *E. coli* was not detected in the studied DM samples [15; 62] or it was detected only in a limited number of samples in low counts (up to 2 log cfu mL^−1^) [59,64,70,71]. Similarly, Enterobacteriaceae and coliforms in DM ranged from 0.68 to 3.2 log cfu mL^−1^ [2,10,23,54,56,60] and from 2.14 to 2.85 log cfu mL^−1^ [12,64,72,73], respectively. The source of these indicators of fecal contamination of raw milk is usually the mat, and the vector of their transmission is usually the surface of the animal’s udder or the worker’s hand [74]. There are no data on the number of coliform bacteria on the udder surface of donkeys, but there are some reports on dairy cows. The number of coliform bacteria on the udder of cows rarely exceeds 100 cells, although they make up the majority of the mat microbiota [74]. This can also be the case with a donkey’s udder. Furthermore, the smaller size of the donkey’s udder and its natural anatomical location limit the exposure of the teats to bacterial contamination [32,75].

Microbiological studies of raw DM, which included enumeration of *B. cereus*, sulfite-reducing Clostridia and aerobic sporogenic bacteria, are very limited. Previous studies of raw DM showed the presence of *B. cereus* ranging from less than 1 to 3.08 log cfu mL^−1^ [54,61,76], while the number of aerobic sporogenic bacteria was 1.3 and 0.23 log cfu mL^−1^ [15,28]. Sulfite-reducing Clostridia and aerobic sporogenic bacteria including *B. cereus* are ubiquitous microorganisms that can be isolated from soil, feed, mats and feces. Contamination of raw milk with these bacteria usually occurs by two routes: direct contamination of the udder through soil and indirect contamination through consumption of contaminated feed [77,78]. The absence of *B. cereus* in the examined DM samples could also be due to the intensive dairy farming in the Special Zasavica Nature Reserve, where donkeys are not allowed to graze during milking.

According to the literature, the number of yeasts and molds in raw DM ranged from 0.69 to 4.50 log cfu mL^−1^ [2,10,28]. Malissiova et al. [62] analyzed 41 raw DM samples originating from Greek and Cypriot farms. All samples were positive for yeasts and molds, with values up to 4.23 log cfu mL^−1^. In contrast to these results, yeasts and molds were below the limit of quantification in all raw DM samples collected from an autochthonous Serbian breed [15]. Contamination of milk with yeasts and molds most commonly occurs through the air or through contact of milk with a contaminated surface of milking equipment. From a public health perspective, the presence of yeasts and molds in raw milk is of minor importance, as pathogenic yeasts such as *Candida albicans* and *Cryptococcus neoformans* are not foodborne pathogens [79], while the carcinogenic aflatoxin M1 can occur in milk if animals are fed on feed contaminated with aflatoxin B1 [80].

### 4.2. Changes in the Microbial Population during Storage

According to the results, the DM was safe for consumption during all six days of storage, as it met the requirements of guidance for producers of raw drinking milk for direct human consumption [81]. Previous studies have shown some variability in the microbiological quality of raw DM during prolonged storage at 4 °C [15,28]. The TBC in raw DM at the beginning of storage was reported to be 4.57 ± 0.29 log cfu mL^−1^ [15] and 4.34 ± 0.37 log cfu mL^−1^ [28] and remained at the same level until the fourth day of storage. The higher initial TBC in previous studies might be related to various hygiene failures during milking and milk manipulation. The DM used in this study (Table 1) was a mixture of nine individual milk samples with a TBC of less than 2 log cfu mL^−1^. Therefore, the initial TBC (Table 1) was lower compared to previously published results [15,28]. If samples with a TBC above 4 log cfu mL^−1^ (Figure 1) were selected and mixed, the results of this study would likely be similar to those previously reported [15,28].

### 4.3. Characterization of the Donkey Milk Proteins

According to the literature, the LZ content in DM ranged from 1 g L^−1^ [82] to 4 g L^−1^ [2]. The different levels of these proteins found in DM could be explained by the effects of the lactation stage of the dairy animals and the different analytical methods applied [82]. Previous studies have shown that LZ was present in Domestic Balkan DM in a range from 0.97 to 3.89 g L^−1^, while the observed concentrations of LF ranged from undetectable levels to 54.3 mg L^−1^ [17,83]. Therefore, the results of this study are consistent with previous studies conducted on Domestic Balkan DM.

### 4.4. Antibacterial Activity of Donkey Milk

*S. aureus* was the least sensitive to the antimicrobial activity of DM. This is consistent with the results of the microbiological quality of raw Domestic Balkan DM (Figure 1), as coagulase-positive staphylococci were the only pathogens isolated from the milk samples tested.

Milk and colostrum contain numerous antimicrobial compounds that have specific and non-specific bacteriostatic or bactericidal properties. Antimicrobial compounds in milk can generally be divided into immunoglobulins and non-immune proteins [84,85]. Immunoglobulins are the main antimicrobial agents of colostrum and therefore represent a kind of passive immunity. Non-immune proteins, including the major iron-binding glycoprotein LF and the enzyme LZ, play a more direct role in inhibiting bacterial invasion. The strong antibacterial activity of DM is primarily attributed to the high concentration of LZ in this milk [2,11,14,28,82,84].

The antibacterial activity of LZ is related to the ability of this protein to hydrolyze β-1,4-glycosidic bonds between N-acetylmuramic acid and N-acetylglucosamine present in the peptidoglycan layer of the bacterial cell wall, leading to its degradation and cell death. According to the literature, Gram-positive bacteria are very sensitive to LZ because their cell wall consists of 90% peptidoglycan [85]. Gram-negative bacteria, on the other hand, are less susceptible to its action because peptidoglycan makes up 5 to 10% of their cell wall structure. Moreover, these bacteria have an outer membrane that does not allow the LZ molecules to enter the target sites in the peptidoglycan structure [85,86].

However, comparing the results of the antibacterial activity of DM on Gram-positive and Gram-negative bacteria obtained in this study, a weaker antibacterial potential of this milk was found against Gram-positive strains (Figure 3). In Gram-positive bacteria, there are various defense mechanisms against the hydrolytic action of LZ including different peptidoglycan modifications such as O-acetylation and N-deacetylation [87,88,89,90,91,92]. The degree of O-acetylation varies depending on the bacterial strain and the age of the culture, and typically 15 to 70% of the peptidoglycan in the cell wall is altered [89,91]. The slowing of growth of *L. monocytogenes* (Figure 3a) and *S. aureus* (Figure 3c) in DM may be due to the fact that the peptidoglycan layer is not completely modified, so that LZ still has some access to target sites on the peptidoglycan.

Previous research has shown that the strong antibacterial activity of DM against Gram-negative bacteria is most likely due to the non-enzymatic mode of action of LZ from DM [17,83] and the synergistic effect of LZ and LF [18]. This non-enzymatic model of LZ action has already been demonstrated in equine LZ [93,94]. The binding of calcium ions reduces the hydrophobicity of equine LZ and improves its antibacterial activity against Gram-negative bacteria [95,96,97,98]. Considering that horses and donkeys are closely related mammalian species belonging to the same family *Equidae* [99], it is very likely that their LZ structures are similar, and that DM LZ also has the ability to bind calcium ions. Accordingly, Šarić et al. [17,83] reported calcium-dependent antibacterial activity of DM against *E. coli*, *S*. Enteritidis and *S*. Typhimurium at 38 °C.

There are no available data on the antibacterial effect of LF from DM against *L. monocytogenes* and *S. aureus*. Previous investigations of the antibacterial activity of DM against *E. coli* [17] and selected *Salmonella* strains [83] conducted at 38 °C have shown no dependence of the antibacterial effect of this milk on the content of LF. Although some contribution of LF to the overall antibacterial activity of DM against tested pathogens should not be discounted, LZ is most likely the main antibacterial agent in DM, since LZ was detected in a much higher concentration than LF.

However, DM represents a very complex medium that contains numerous compounds that can contribute to its overall antibacterial activity. Šarić et al. [100] reported a high proportion of fatty acids with known antibacterial activity toward Gram-positive bacteria, such as linoleic, lauric and oleic acids, in the total fatty acid content of the Domestic Balkan DM. Furthermore, LZ can acts in synergy with some other milk compounds including lacto-peroxidase, N-acetyl-b-D-gluco-aminidase, immunoglobulins and fatty acids [11,101]. Therefore, further research is needed to fully elucidate the antibacterial mechanism of donkey milk.

## 5. Conclusions

More than 97% of the DM samples tested meet the requirements laid down for raw drinking milk for direct human consumption [81] since TBC was slightly above the maximum permissible limit in only four tested samples. Absence of pathogens tested and low levels of TBC, LAB and CPS in DM indicated good hygiene practice during milking. The microbial quality of DM during prolonged storage indicated the DM was safe for consumption during the six-day storage [81]. DM showed antibacterial potential against all tested pathogens, with the strongest antibacterial activity against *E. coli*. *S. aureus* proved to be the most resistant to this antibacterial activity.

Although the Serbian regulation on the quality of raw milk [36] mentions donkey milk, it does not contain quality criteria for raw drinking milk intended for direct human consumption. It only prescribes the microbiological criterion (1.5 million cfu mL^−1^ of microorganisms) for classifying goat, sheep and other domestic animals’ milk into the first and second quality classes. This criterion is based on the microbiological quality of raw goat and raw sheep milk and is not suitable for raw DM. Therefore, this study provided useful data for the laying down of recent national regulation for the production and commercialization of raw DM. Currently, primary producers must ensure that raw milk is marketed directly to the final consumer no later than 24 h after the milk has been milked and cooled to 4 °C. Given the excellent microbiological quality of DM and its great antibacterial potential, consideration should be given to extending the storage time of raw DM at 4 °C, during which it can be safely consumed. Before doing so, all the requirements of good manufacturing and hygiene practices must be met.

## Figures and Tables

**Figure 1 animals-13-00327-f001:**
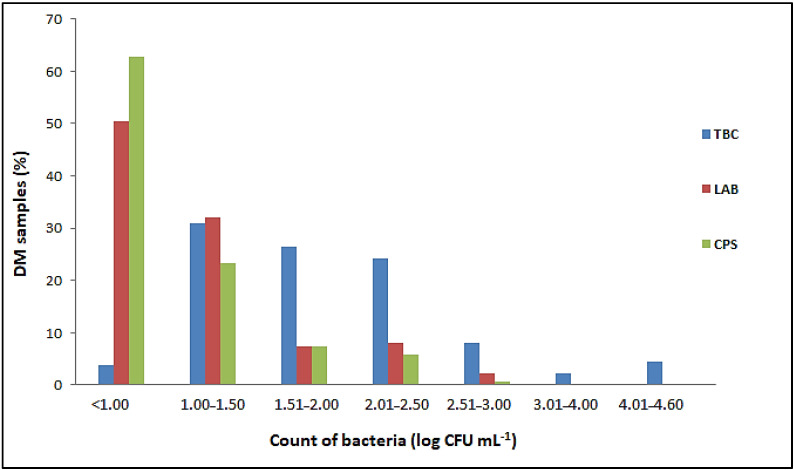
Proportion of DM samples with different ranges of bacterial counts.

**Figure 2 animals-13-00327-f002:**
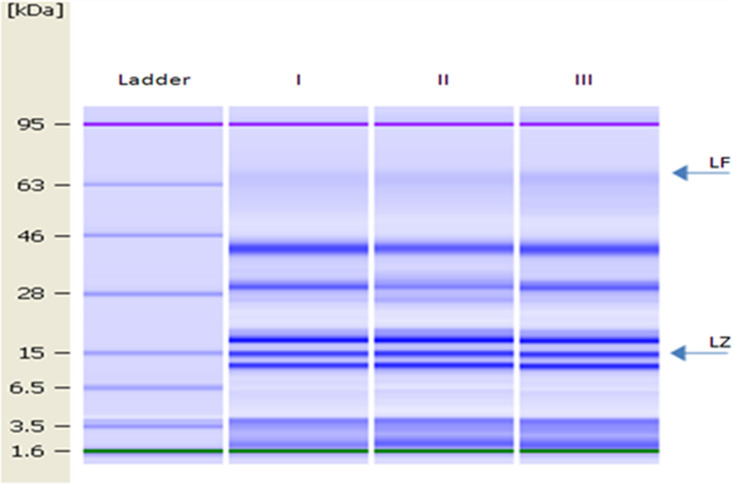
Lab-on-a-Chip gel images of lane 1, molecular mass ladder (1.6, 3.5, 6.5, 15, 28, 46, 63 and 95 kDa); lanes I–III, DM samples. Arrows indicate the position of LF and LZ.

**Figure 3 animals-13-00327-f003:**
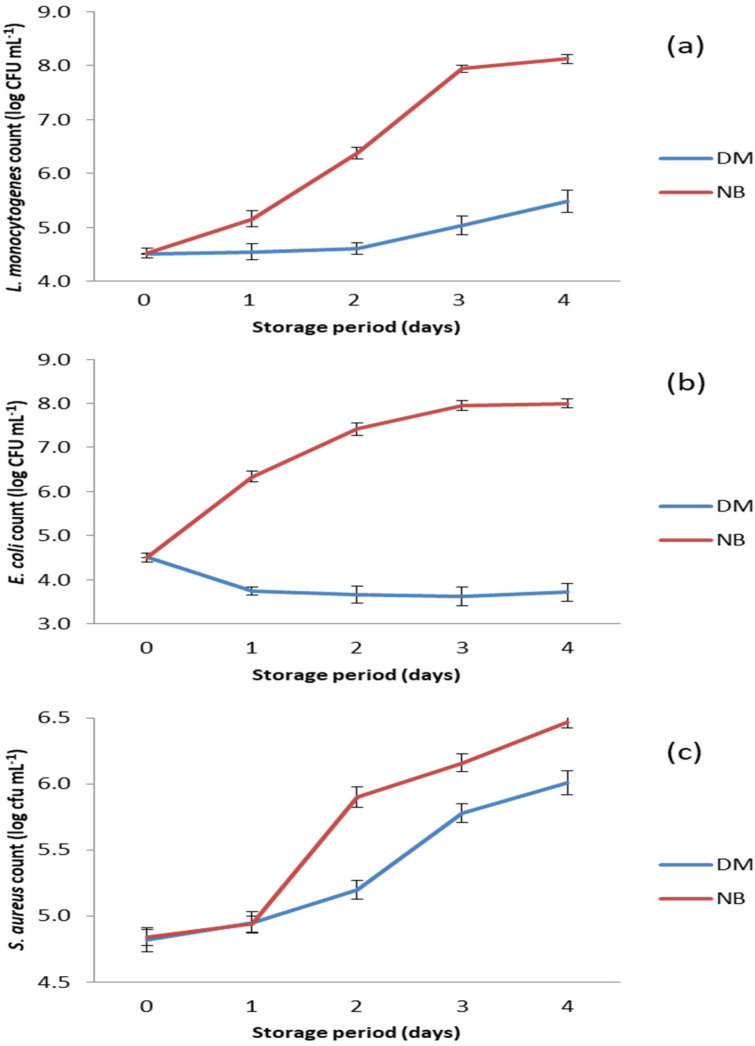
Counts of bacteria tested in the DM and the nutrient broth (NB) during storage at 15 °C. (**a**) *S. aureus* count; (**b**) *E. coli* count; and (**c**) *L. monocytogenens* count.

**Table 1 animals-13-00327-t001:** Microbial quality of donkeys’ milk during storage at 4 °C.

	Storage Period (Days)						
	0	1	2	3	4	5	6
TBC	1.72 ^a^ (0.54)	1.77 (0.48)	1.77 ^a^ (0.34)	1.70 ^а^ (0.65)	1.75 ^а^ (0.38)	1.77 ^а^ (0.45)	1.78 ^а^ (0.38)
LAB	0.97 ^a^ (0.31)	0.98 ^a^ (0.30)	1.11 ^a^ (0.25)	1.21 ^a^ (0.19)	1.26 ^a^ (0.19)	1.24 ^a^ (0.07)	1.34 ^a^ (0.09)
ASB	<1	<1	<1	<1	<1	<1	<1
ENT	<1	<1	<1	<1	<1	<1	<1
COL	<1	<1	<1	<1	<1	<1	<1

Results are expressed in log cfu mL^−1^. Each value is the mean of three independent experiments. Standard deviation values are given in parentheses. Means with different superscript letters in the same row are significantly different (*p* < 0.05). Abbreviations: TBC—total bacterial count; LAB—lactic acid bacteria; ASB—aerobic sporogenic bacteria; ЕNТ—Enterobacteriaceae; COL—coliforms.

## Data Availability

Not applicable.

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
