# Peer review of "Microbiological Quality of Raw Donkey Milk from Serbia and Its Antibacterial Properties at Pre-Cooling Temperature"

_animals, 2023, doi:10.3390/ani13030327_

Round 1

Reviewer 1 Report

Title: Microbiological Quality of Raw Donkey Milk from Serbia and Its Antibacterial Properties at Pre-cooling Temperature

 General comments:

The issue reviewed is relevant and interesting, but the document presents serious problems.

 Specific comments:

The term “microflora” is outdated and must be replaced by “microbiota” along the document (abstract included).

Why was a simple abstract included? It is misleading, since it does not contain all the data necessary to consubstantiated the conclusions drawn.

Furthermore, the abstract is a crucial section of every manuscript and must follow the ideas: introduction/contextualization and major aims, methodology overview, major results, most important conclusions. Please revise accordingly.

Material and methods

Every morning before milking, the udders were washed under cold running water and then dried with a towel. The individual milk samples were collected in sterile bottles and stored in an ice box at 4 °C during transport to the laboratory.” Is this normal procedure, or was it only performed like this for this specific study?

What are the microbial/quality control standards for raw milk samples in Serbia?

2.4. Determination of the content of LZ and LF” abbreviations should not be used in titles

The antibacterial assay is confusing. Start by inoculum preparation, before the description of the actual assay and describe it thoroughly, so it may be repeated by other researchers.

Results

I believe that results display in Table 1 are not the best form to present bacterial counts. Maybe a graphic representation (box plots), instead of a Table.

Why are not ALL microbial counts included in the Table (example: yeast and molds)

Where is the comparison between bacterial counts found in this study and the microbial load accepted by legislation for raw milk?

Regarding Table 2, once again a graphic representation would be preferable.

Figures 2-4 should be combined in only one figure.

Bacteria scientific names must be in italics along the document.

It is difficult to perform a reliable comparison between microbial growth in Nutrient broth and Donkey milk. Couldn’t you have sterilized the milk and use it for comparison?

 Discussion

As the CPS was detected in low numbers (Table 1), these pathogens were not capable of producing large amounts of enterotoxins required to cause foodborne poisoning.” This statement is incorrect, the authors did not test enterotoxin’s production…

Information on legislation requirements was not included nor discussed.

As aforementioned comparion between growth in nutrient broth and NA is ambiguous, this bias must be properly discussed to avoid missinterpretations.

Author Response

-

Reviewer 2 Report

The authors have provided some interesting information on the microbiological safety of a less frequently studied non-cow milk that is not commonly consumed by the majority population. Furthermore, an antibacterial potential of donkey milk was reported, though all the associated component(s) were not identified. Nevertheless, clarifications in the materials and methods part are required to better understand the experimental design of this study as well as an extensive editing of the discussion to reduce its length to avoid repetition and provide only the information that is directly linked to the presented findings. Specific comments are as follows. 

1.       Line 33-34. For which time points it was investigated?

2.       Line 41. It might be better to change ‘six days’ to ‘the six-day’.

3.       Line 50. Please remove ‘the’ before ‘global’.

4.       Line 62. Please remove ‘The’.

5.       Line 108. Could you please provide a range of repeated samplings per individual animal? E.g.  what was the minimum number of milk samples per animal?

6.       Line 113. Was a teat disinfectant not used? Were the first squirts of milk discarded as done in standard milk sampling practice?

7.       Line 115. It doesn’t mention whetder the samples are cultivated immediately or whether they are stored  (for how long and under which conditions )  

8.       Line 120. Please add ‘according to the respective ISO methodology’ at the end of the sentence as done in line 125.

9.       Line 122. Please make sure that the information on the companies producing the media, consumables etc is provided in detail, when first mentioned.

10.   Line 127. Were these nine samples a subset of the 137 samples? If so, which criteria were used to determine this number and which samples will be pooled for the evaluation of the microbial population during storage?

11.   Line 129. As comment 2.

12.   Line 130-137. This is a repetition of section 2.2. Please just state the bacterial groups monitored and say that enumeration was performed using the above-mentioned methodologies.

13.   Line 152-153. If this is the same bulk milk as in section 2.3, it is not necessary to repeat how it was prepared again.  Just start the paragraph by saying ‘The three subsamples of the bulk DM were….’.

14.   Line 161. On which criteria was the temperature of 15 oC selected as a storage condition?

15.   Line 175. The word ‘Clostridia’ does not need to be in italics.

16.   Line 188. It might be better to change ‘the’ to ‘a’.

17.   Line 193. Correct p-value, if not significantly different (P>0.05).

18.   Line 195-197. Please italicise bacterial genera and species.

19.   Line 204. Shouldn’t LF be roughly in the middle of the 63 and 95 kDa in Figure 1 if its size is 78.1 kDa? Why is it so close to the 63kDa ladder?

20.   Could you please include the standard deviation in each time point for DM and NB and maybe stars where p<0.05 in Figures 2-4? Also, italicise bacterial names.

21.    Line 216. Include p-value, if statistically significant.

22.   Line 220, 222, L. monocytogenes should be in italics.

23.   Line 223. Please include p-value.

24.   Line 227, 230, 232. S. aureus should be in italics.

25.   Line 230. Please include p-value.

26.   Line 237, 238. E. coli should be in italics.

27.   Line 238, 240. Please include p-value.

28.   Line 251-271. Please combine these two paragraphs into a single one to avoid repetition.

29.   Line 300-301. This part should be rewritten. Either move these bacterial groups at the beginning of the sentence, together with E. coli or include them in a new sentence.

30.   Line 307-308. This sentence does not seem correct. Why cannot E coli survive on the skin of the udder for a long time? Competition with skin microbiota? Commonly present in low numbers?

31.   Line 319. Please change ‘throw’ to ‘excrete’.

32.   Line 327. Was this measured in this study? If not, please provide with a reference.

33.   Please shorten section 4.1. It’s two pages long and quite repetitive. Please try grouping them, where possible, rather than discussing each bacterial group individually.

34.   Line 375. Please remove ‘of our study’.

35.   Line 379-380. These findings are indicative of good milking hygiene (as pointed out in section 4.1) and good storage practices rather than antibacterial activity of milk.

36.   Line 389-390. Please include a reference.

37.   Please shorten section 4.4. Perhaps don’t provide such analytical descriptions on the antibacterial mechanisms of the enzymes, bit only focus on the main points of these mechanisms relevant to the findings of the presented study.

38.   Line 471. The absence of pathogens and the low bacterial counts are not necessary indicative of DM antibacterial activity. Please remove this statement.

Author Response

-

Round 2

Reviewer 1 Report

Thank for adressing my comments. However, I still believe that the abstract's section can be improved. The ideas are not well organized, which difficults reading and comprehension. Please try to simplify and clarify the text.

Figures 1-3 require further information on the legend (the numbers on the right correspond to log cfu mL-1, this information is on the text but not on the figure itself). Aditionally, in my opinion a graphic format other than "pie" might work better

Please revise "count" to "counts" throughout the manuscirpt

Author Response

Response to Reviewer 1 Comments

AUTHORS: We want to thank the Reviewer for his/her thorough reading of our paper and all the useful comments and corrections which have been very helpful in improving the manuscript. We hope that the following answers will clarify the requested points.

Point 1: However, I still believe that the abstract's section can be improved. The ideas are not well organized, which difficults reading and comprehension. Please try to simplify and clarify the text.

Response 1: The abstract has been revised.

Point 2: Figures 1-3 require further information on the legend (the numbers on the right correspond to log cfu mL-1, this information is on the text but not on the figure itself). Aditionally, in my opinion a graphic format other than "pie" might work better

Response 2: "pies" have been replaced with a single column chart (Figure 1 in the revised manuscript)

Point 3: Please revise "count" to "counts" throughout the manuscript

Response 3: "count" has been revised to "counts" throughout the manuscript.

Reviewer 2 Report

The authors have adequately addressed the comments and questions from the first revision round. The manuscript has been substantially improved, although I reckon that some additional editing in the discussion is required. Specific comments are as follows. 

1.      Line 20. Please remove statement ‘and great antimicrobial potential of donkey milk’ and start the next sentence with it e.g. ‘Regarding the antimicrobial potential of donkey milk, E. coli appeared to be the most 20 sensitive to the antibacterial activity of donkey milk….’

2.      Line 28. Please change ‘population’ to ‘populations’.

3.      Line 69. Please change ‘convictions’ to ‘beliefs’.

4.      Line 104. Please change ‘a’ to ‘the’.

5.      Line 117. Please add ‘according to the respective ISO methodology’ at the end of the sentence.

6.      Line 119, 121,163. Please include the cities in the information on the companies producing the media.

7.      Line 121. Please add ‘according to’ before ISO.

8.      Line 138-139. Please correct ‘BioanaLZer’ to ‘Bioanalyzer’.

9.      Line 150. Please change ‘test’ to ‘tested’.

10.   Line 151. Please change ‘shaken’ to ‘mixed’.

11.   Line 155. Please align the title with the other titles.

12.   Line 157. Please add ‘the’ after ‘with’.

13.   Line 184, 192, 203. Please rephrase the captions of Figures 1-3. It is not clear that the pie chart shows the percentage of DM samples with different log cfu ml-1 for each bacterial group.

14.   Line 195. Please change ‘A’ to ‘The’.

15.   Line 205. It might be better to change ‘the’ to ‘a’.

16.   Line 212-213. Please italicise L. monocytogenes.

17.   Line 223-224. Maybe change ‘lane 2’ to ‘lanes I-III’ to be more evident that each lane is a subsample of the bulk DM.

18.   Figure 5 is more complicated than Figures 2-4 in the previous version of the manuscript. Perhaps it might be better to include this figures side by side e.g. Figure 5 a) L. monocytogenes, b) E. coli and c) S. aureus. Also, could you please include the standard deviation in each time point for DM and NB and maybe stars where p<0.05?

19.   Line 249-250. Please include p-value.

20.   Line 257. Please remove ‘milk’.

21.   Line 262-264. Rephrase this sentence. E.g. ….showed that the high TBC determined in raw DM was associated with hygiene deficiencies….. 

22.   Line 313. Please remove ‘a’.

23.   Line 319. Please remove ‘sources’.

24.   Line 331-338. It is said that the seasonal variation on the microbial quality was not investigated, but is then stated that season did not influence the TBC, LAB and CPS levels without the relevant data being presented in the Results section. Perhaps remove this paragraph to avoid the confusion.

25.   Line 353. Please add ‘the’ after ‘to’ and remove ‘sources’.

26.   Line 365. It might be better to change ‘like’ to ‘and’.

27.   Line 383. These figures are not included in the revised manuscript. Please add the correct figures.

28.   Line 381-393. Combine them into a single paragraph.

29.   Line 423-424. Perhaps this sentence is not needed and can be removed.

Author Response

Response to Reviewer 2 Comments

AUTHORS: We want to thank the Reviewer for his/her thorough reading of our paper and all the useful comments and corrections which have been very helpful in improving the manuscript. We hope that the following answers will clarify the requested points.

The authors have adequately addressed the comments and questions from the first revision round. The manuscript has been substantially improved, although I reckon that some additional editing in the discussion is required. Specific comments are as follows.

Point 1: Line 20. Please remove statement ‘and great antimicrobial potential of donkey milk’ and start the next sentence with it e.g. ‘Regarding the antimicrobial potential of donkey milk, E. coli appeared to be the most 20 sensitive to the antibacterial activity of donkey milk….’

Response 1: The statement ‘and great antimicrobial potential of donkey milk’ has been removed, and the next sentence is started with ‘Regarding the antimicrobial potential of donkey milk,…

Point 2: Line 28. Please change ‘population’ to ‘populations’.

Response 2: ‘population’ has been changed to ‘populations’.

Point 3: Line 69. Please change ‘convictions’ to ‘beliefs’.

Response 3: ‘convictions’ has been changed to ‘beliefs’.

Point 4: Line 104. Please change ‘a’ to ‘the’.

Response 4: ‘a’ has been changed to ‘the’

Point 5: Line 117. Please add ‘according to the respective ISO methodology’ at the end of the sentence.

Response 5: ‘according to the respective ISO methodology’ has been added.

Point 6: Line 119, 121,163. Please include the cities in the information on the companies producing the media.

Response 6: The names of the cities have been included in the information on the companies producing the media.

Point 7: Line 121. Please add ‘according to’ before ISO.

Response 7: ‘according to’ has been added before ISO.

Point 8: Line 138-139. Please correct ‘BioanaLZer’ to ‘Bioanalyzer’.

Response 8: ‘BioanaLZer’ has been corrected to ‘Bioanalyzer’

Point 9: Line 150. Please change ‘test’ to ‘tested’.

Response 9: ‘test’ has been changed to ‘tested’

Point 10: Line 151. Please change ‘shaken’ to ‘mixed’.

Response 10: ‘shaken’ has been changed to ‘mixed’

Point 11: Line 155. Please align the title with the other titles.

Response 11: The title ‘2.5.2 Test at 15 °C’ has been aligned with the other titles

Point 12: Line 157. Please add ‘the’ after ‘with’.

Response 12:  ‘the’ has been added after ‘with’.

Point 13: Line 184, 192, 203. Please rephrase the captions of Figures 1-3. It is not clear that the pie chart shows the percentage of DM samples with different log cfu ml-1 for each bacterial group.

Response 13: "pies 1-3" have been replaced with a column chart (Figure 1). The caption of Figure 1 has been rephrased according to your recommendations.

Point 14: Line 195. Please change ‘A’ to ‘The’.

Response 14: ‘A’ has been changed to ‘The’

Point 15: Line 205. It might be better to change ‘the’ to ‘a’.

Response 15: ‘The’ has been changed to ‘A’

Point 16: Line 212-213. Please italicise L. monocytogenes.

Response 16: L. monocytogenes has been italicised.

Point 17: Line 223-224. Maybe change ‘lane 2’ to ‘lanes I-III’ to be more evident that each lane is a subsample of the bulk DM.

Response 17: ‘lane 2’ has been changed to ‘lanes I-III’

Point 18: Figure 5 is more complicated than Figures 2-4 in the previous version of the manuscript. Perhaps it might be better to include this figures side by side e.g. Figure 5 a) L. monocytogenes, b) E. coli and c) S. aureus. Also, could you please include the standard deviation in each time point for DM and NB and maybe stars where p<0.05?  

Response 18: These figures have been included in only one figure (Figure 3 in the revised manuscript) according to your recommendations. The standard deviation in each time point for DM and NB has also been included.

Point 19: Line 249-250. Please include p-value.

Response 19: p-value has been included

Point 20: Line 257. Please remove ‘milk’.

Response 20: ‘milk’ has been removed

Point 21: Line 262-264. Rephrase this sentence. E.g. ….showed that the high TBC determined in raw DM was associated with hygiene deficiencies…..

Response 21: Mentioned sentence has been rephrased.

Point 22: Line 313. Please remove ‘a’.

Response 22: ‘a’ has been removed.

Point 23: Line 319. Please remove ‘sources’.

Response 23: ‘sources’ have been removed.

Point 24: Line 331-338. It is said that the seasonal variation on the microbial quality was not investigated, but is then stated that season did not influence the TBC, LAB and CPS levels without the relevant data being presented in the Results section. Perhaps remove this paragraph to avoid the confusion.

Response 24: This paragraph has been removed.

Point 25: Line 353. Please add ‘the’ after ‘to’ and remove ‘sources’.

Response 25: ‘the’ has been added after ‘to’ and ‘sources’ have been removed

Point 26: Line 365. It might be better to change ‘like’ to ‘and’.

Response 26: ‘like’ has been changed to ‘and’

Point 27: Line 383. These figures are not included in the revised manuscript. Please add the correct figures.

Response 27: The correct figure (Figure 5) has been added.

Point 28: Line 381-393. Combine them into a single paragraph.

Response 28: The mentioned two paragraphs have been combined into one paragraph.

Point 29: Line 423-424. Perhaps this sentence is not needed and can be removed.

Response 29: This sentence has been removed.